# Dissecting Seed Proanthocyandin Composition and Accumulation under Different Berry Ripening Process in Wine Grapes

**Aoyi Liu** [1,2]**, Jingjing Wang** [1,2]**, Xuechen Yao** [1,2]**, Nongyu Xia** [1,2]**, Qi Sun** [1,2]**, Changqing Duan** [1,2]
**and Qiuhong Pan** [1,2,*]

1    Center for Viticulture & Enology, College of Food Science and Nutritional Engineering,
     China Agricultural University, Beijing 100083, China
2    Key Laboratory of Viticulture and Enology, Ministry of Agricultural and Rural Affairs,
     Beijing 100083, China
*    Correspondence: panqh@cau.edu.cn; Tel.: +86-10-627-361-91

**Abstract:** Grape berry proanthocyandin (PA) mainly exists in the skin and seeds. Its content and composition determine the intensity of bitterness and astringency. Affected by global warming, the world's wine-producing regions, in particular in dry-hot regions such as western China, are facing the problem of unsynchronized berry ripening and seed ripening. Therefore, it is urgent to understand the influence of berry ripening progression on the composition and accumulation of seed PA, ultimately providing strategies for grape harvest decisions. In this paper, *Vitis vinfera* L. cv. Cabernet sauvignon and Marselan grapes from four sub-regions with different maturation processes were used as experimental materials to study the changes of soluble and insoluble PA contents as well as differences in their composition and mean degree of polymers (mDP) in seeds. The results showed that compared with 'Cabernet sauvignon' seeds, the mDP of soluble and insoluble PA were higher in 'Marselan' seeds. Both varieties showed that the grape berry, with the fastest sugar accumulation, had relatively high soluble PA content in seeds and a high content of (-)-epigallocatechin-3-gallate and (-)-epicatechin in the seed PA composition units. In contrast, the 'Cabernet sauvignon' grapes from the YQ vineyard exhibited the slowest sugar accumulation speed among the four studied vineyards, and their seed PA had the highest mDP and the lowest proportion of (-)-epigallocatechin-3-gallate in the composition units when commercially harvested. According to the results, it is suggested that a faster maturation process would bring about higher levels of bitterness composition, such as (-)-epigallocatechin-3-gallate in seed PA, which is not conducive to the formation of good-tasting tannins.

**Keywords:** soluble PA; insoluble PA; ripening progression; grape seed

## 1. Introduction

Proanthocyandin (PA), also known as condensed tannin (CT), is a polyphenolic compound that widely exists in plants [1,2]. In grapes, it is mainly found in the skin, stems and seeds. There are five flavan-3-ol monomers commonly found in grapes, which are: (2R,3S)-2,3-trans-(+)-Catechin (C), (2R,3R)-2,3-cis-(-)-Epicatechin (EC), (2R,3R)-2,3-cis-(-)-Epicatechin-3-gallate (ECG), (2R,3R)-2,3-cis-(-)-Epigallocatechin (EGC), and (2R,3R)-2,3-cis-(-)-Epigallocatechin-3-gallate (EGCG) [3]. These monomers are polymerized to form PA, which contributes to the intensity of bitterness and astringency of grape berries and wine. The mean degree of polymerization (mDP) is an index used to measure the number of monomers in PA composition. It is generally believed that with an increase in the mDP, the water solubility of PA decreases. When the solubility is too low, precipitation will be formed. The astringency intensity of polymeric PA has a significant correlation with the mDP, the rate of gallation and the total concentration of PA [4]. In general, the larger the

mDP is, the stronger the astringency will be [5]. By comparison, flavan-3-ol monomer usually presents bitterness. Quantitative analyses of the bitterness and astringency of Cs from green tea indicated that the astringency of galloylated flavan-3-ols is much stronger than that of non-galloylated flavan-3-ols, which explains why ECG is the strongest astringency monomer in grape berry's PA [6]. Therefore, with the increase in ECG in the composition units of PA, both astringency and bitterness intensity will be enhanced [7]. Researchers found that PA content and its composition had an obvious impact on the quality of wine, not only contributing to the astringency and bitterness of wine, but also affecting the coloration of the wine [8].

Soluble PA is usually defined to be the PA component that can be extracted with 70% acetone/water solution, and insoluble PA is the PA component in the residual residue [9]. The previous study found that there is 54% extractable PA in skin, 30% in seed and 15% in flesh, while only 25% of the grape berry's total tannins can be extracted into wine in the winemaking process [10]. Given that phenolic compounds in seeds account for 20–55% of the total polyphenol in whole grape berries, it is commonly considered that the content of soluble PA from seeds is higher than that of skin in the resultant wine [5]. Not only that, seed PA has a more positive effect in stabilizing the color of wine because the combination of flavanols and anthocyanins can prevent anthocyanin oxidation [11,12]. Therefore, studying the content and composition of PA in grape seeds and its changes in the ripening process will help winemakers to reasonably predict the maturity of tannins when grapes are harvested.

Seed PA is mainly composed of C, EC and ECG [12]. Its mDP is around 2.3–30.3 and has a higher galloylation rate [5]. At present, studies have shown that there is a relationship between berry growth and seed growth, and the process of berry growing usually takes sugar accumulation as the reference. Regarding the changes in seed PA during berry maturation, it is generally believed that seed PA is synthesized and accumulated in the first growth stage of fruit development, and its content decreases in the mature stage [5]. Kennedy et al. [13] proposed that the changes of *Vitis vinifera*. L. cv. Syrah grape seed polyphenols correspond to four stages of berry growth: from flowering to the first month of fruit development, PA synthesis and accumulation starts in the seed; before the berries begin to color, the composition units of PA, flavan-3-ol monomers, accumulate, but the formation of PA is not synchronized with monomer synthesis; from the beginning of berry véraison to the maximum berry weight, the water content of the seeds decreases, the PA content of the seeds increases, and its oxidation occurs; finally, the extractable PA has a slight change in composition and content, and non-programmed oxidation happens. Some researchers have also merged the first two stages and put forward a three-stage model [14]. The above studies have demonstrated a potential relationship between berry ripening and seed ripening, which can be used to manage and control the composition and content of tannins in seeds at harvest.

Global warming has become a widespread concern, and the global average temperature in 2021 was about $1.11 \pm 0.13$ °C higher than that in 1850–1900, before industrialization [15]. According to the sixth assessment report of the Intergovernmental Panel on Climate Change (IPCC), the surface temperature in the next two decades is likely to be 0.4–1.0 °C higher than that in 1995–2014 [16]. The increase in global temperatures has led to earlier grape phenology and faster ripening speed [17], and this change has been particularly obvious in the last 10–30 years. The recorded data have shown that the harvest time of grapes has advanced by about 2–3 weeks in the past 100 years [18–20]. The eastern foothills of the Helan Mountains are an important wine-producing area in China, with a large temperature difference between day and night, abundant sunlight and long hours of sunshine, resulting in fast ripening progression and high sugar accumulation of grape berries. In recent decades, the average air temperature during grapevine growth season at the eastern foothills of the Helan Mountains has risen 0.29–0.89 °C per decade from 1981 to 2015 [21], which has had significant impact on the quality of the wine grapes produced. At the same time, the north of the Helan Mountains is higher than the south, forming a

few quite microclimate sub-regions from north to south and thus resulting in an unsynchronized ripening progression of the berries between the various sub-regions. However, there are still few research studies on the differences in the ripening process of berries in the sub-regions of this Chinese wine production region and their impact on the quality of wine grapes.

Most previous studies have mainly been concerned about the influence of a particular terroir element, such as soil conditions, mesoclimate, altitude, drought, etc., on the phenolics of the grapes and thus on the typical quality of the wine [22–24]. Little attention has been paid to the influence of terroir on grape tannins, in particular seed tannins. In this paper, we used the seeds of 'Cabernet sauvignon' and 'Marselan' grapes with different ripening processes from four sub-regions at the eastern foothills of the Helan Mountains as the experimental materials, aimed to explore the relationship between the berries' sugar accumulation and seed PA content, mDP or composition. The research findings will provide strategies for synergistic grape berry and seed ripening processes and for the production of high-quality wine materials.

## 2. Materials and Methods

### 2.1. Experimental Materials

2.1.1. Information of Sub-Regions

This study was carried out in 2021 in four sub-regions at the eastern foot of the Helan Mountain, a well-known wine production area in Ningxia, China. The sub-regions are Zhihui Vineyard (ZH) (106.12° E, 38.28° N), Chateau Yuquan (YQ) (106.05° E, 38.27° N), Xige Estate (XG) (105.89° E, 38.07° N) and Hongbao Vineyard (HB) (105.52° E, 38.30° N). The geographical locations of the four sub-regions are exhibited in Figure 1. In the same sub-regions, *Vitis vinifera* L. cv. Cabernet Sauvignon vineyard and *V. vinifera*. L. cv. Marselan vineyard with adjacent plots and similar soil type were selected as sampling sites. All vines of the two varieties were self-rooted. The soil types, soil physical–chemical index [25–27] and grape vine planting parameters are shown in Table 1.

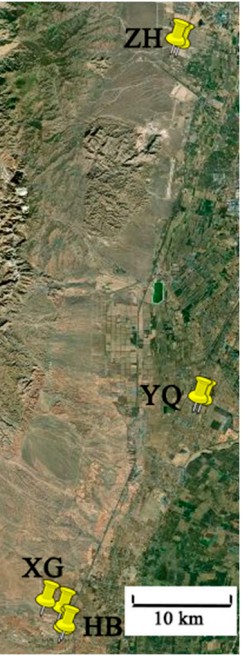

**Figure 1.** The geographical locations of the four sub-regions.

**Table 1.** Soil type and their physical–chemical parameters of sub-regions and description of grapevines.

| Sub-Regions | Soil Type | pH | Organic Content g/kg | Soil Conditions Available Nitrogen mg/kg | Available Phosphorus mg/kg | Available Kalium mg/kg | 'Cabernet Sauvignon' Grapevine Information | 'Marselan' Grapevine Information |
|---|---|---|---|---|---|---|---|---|
| ZH | Sandy loam | 7.84 | 16.40 | 82.00 | 15.00 | 220.00 | 6.35 ha, planted in 2013, spaced at 1.2 × 4 m (vine × row), orientated north–south rows, Vertical Shoot Positioning, yield 1.99 kg per extension meter. | 6.4 ha, planted in 2015, spaced at 2 × 4 m (vine × row), orientated north–south rows, Vertical Shoot Positioning, yield 1.77 kg per extension meter. |
| YQ | Sandy light sierozem | 8.66 | 8.58 | 9.55 | 27.16 | 104.39 | 2 ha, planted in 2012, spaced at 0.5 × 3.5 m (vine × row), orientated north–south rows, vertical shoot positioning, yield 2.00 kg per extension meter. | 4.73 ha, planted in 2018, spaced at 0.6 × 3.5 m (vine × row), orientated north–south rows, vertical shoot positioning, yield 2.19 kg per extension meter. |
| XG | Aeolian soil | 8.58 | 10.83 | 14.86 | 51.66 | 155.31 | 2.5 ha, planted in 2017, spaced at 1 × 3.5 m (vine × row), orientated north–south rows, inclined single cordon trellis system, yield 1.32 kg per extension meter. | 2 ha, planted in 2017, spaced at 1 × 3 m (vine × row), orientated north–south rows, Vertical Shoot Positioning, yielding 1.51 kg per extension meter. |
| HB | Newly accumulated light sierozem | 8.49 | 6.22 | 15.10 | 28.00 | 147.00 | 1.33 ha, planted in 2010, spaced at 0.4 × 3 m (vine × row), orientated north–south rows, single cordon trellis system, yield 3.47 kg per extension meter. | 2.13 ha, planted in 2017, spaced at 0.4 × 3.2 m (vine × row), orientated north–south rows, single cordon trellis system, yield 4.9 kg per extension meter. |

### 2.1.2. Sample Collection

According to the principle of randomized block design, three biological plots were set up in each vineyard as biological replicates. The biological plots consisted of 10 rows, spaced 40 m apart. Three rows of vines were selected from west to east in each biological plot, and 5 grapevines with the same growth trend were selected from the front, middle and back of each row. In total, each biological plot had 15 vines.

A total of 300 berries were collected from each biological plot. All berries were randomly selected from the upper, middle, and lower parts of the grape cluster and the shady and sunny sides of each row. Sampling of each plot was completed before 10:00 on the same day. After each sampling, grape samples were immediately put into the ice box and transported to the laboratory within one hour. Of the samples from each biological plot, 50 fresh berries were used for the measurement of total soluble solid (TSS) and titratable acid (TA), and the rest were rapidly frozen in liquid nitrogen and stored in a −80 °C refrigerator.

This study focused on investigating the accumulation of seed PA after full coloration, which was used as the first sampling point. Thereafter, the berries were collected every 10 days until commercial harvest. The dates of full coloration of 'Cabernet sauvignon' and 'Marselan' grapes in the four sub-regions were different. Thus, the dates of arrival at each sampling point varied considerably (Table 2).

**Table 2.** Sampling schedule for 'Cabernet sauvignon' and 'Marselan' grapes.

| Date after Full Coloration (DAFC) | | Sub-Regions | | | |
|---|---|---|---|---|---|
| | | ZH | YQ | XG | HB |
| DAFC0 | Cabernet sauvignon | 8/10 | 8/12 | 8/5 | 8/14 |
| | Marselan | 8/5 | 8/7 | 8/5 | 8/14 |
| DAFC10 | Cabernet sauvignon | 8/20 | 8/22 | 8/14 | 8/24 |
| | Marselan | 8/15 | 8/16 | 8/14 | 8/24 |
| DAFC20 | Cabernet sauvignon | 8/30 | 9/2 | 8/24 | 9/3 |
| | Marselan | 8/25 | 8/27 | 8/24 | 9/3 |
| DAFC30 | Cabernet sauvignon | 9/9 | 9/12 | 9/3 | 9/13 |
| | Marselan | 9/4 | 9/7 | 9/3 | 9/13 |
| DAFC40 | Cabernet sauvignon | 9/19 | 9/26 | 9/13 | / |
| | Marselan | / | 9/17 | 9/13 | / |
| DAFC50 | Cabernet sauvignon | 9/29 | 10/1 | / | / |
| | Marselan | / | 9/26 | / | / |
| Commercial harvest | Cabernet sauvignon | 9/29 | 10/1 | 9/13 | 9/13 |
| | Marselan | 9/8 | 9/26 | 9/13 | 9/13 |

Note: Figures in table are dates (m/d); DAFC represents days after berries' full coloration (100% of véraison); Marker '/' represents no sampling. In the last two lines, the commercial harvest time for the two varieties in different wineries is summarized.

*2.2. Experimental Methods*

2.2.1. Determination of Berry TSS and TA

The total soluble solid (TSS) of grapes was determined by a hand-held Brix refractometer. Titratable acid (TA) was titrated by calibrated 0.05 mol/L NaOH solution, and the results were expressed as tartaric acid (g/L) [28].

2.2.2. Extraction and Determination of Seed PA

The extraction and determination of seed PA was carried out by reference to the method of Yu et al. [29] with some modifications. In this study, soluble PA and insoluble PA in the seeds were analyzed separately, including their respective content and mDP and the content of each constituent unit.

**Extraction of soluble PA:** The seeds were artificially picked from the grape berries under liquid nitrogen protection. Twenty seeds were placed into the stainless steel mortar that was pre-cooled by liquid nitrogen, and then were rapidly and fully pestled into powder in the freezing high-speed grinder (MM 400, Retsch, Haan, Germany). Subsequently, 100 mg of seed powder was weighed and fully mixed with 1 mL PA extract (70% acetone/water, 0.1% acetic acid). The mixture was sonicated for 30 min in ice water in darkness. After that, the mixture was centrifuged at $12,000 \times g$ for 10 min at 4 °C, the supernatant was collected, the precipitate was re-extracted twice using the above PA extract solution, and the pooled supernatant was taken as soluble PA crude extract. The precipitate was used for the determination of insoluble PA.

Further extraction was performed for soluble PA. Firstly, the crude extract of soluble PA was fully mixed with an equal volume of chloroform and then centrifuged at $12,000 \times g$ for 5 min at 4 °C. The lower organic phase was discarded, and the upper was collected. The procedure above was repeated twice. Afterward, the pooled upper solution was mixed with an equal volume of *n*-hexane and fully vortexed, followed by centrifugation at $12,000 \times g$ for 5 min at 4 °C. The supernatant was discarded. The procedure above

was repeated twice. Finally, the lower layer of solution was used for the determination of soluble PA.

**Determination of Soluble PA:** First, 2 μL of soluble PA extract and equal volumes of 0, 0.75, 1, 1.5, 2, and 3 mg/mL (-)-epicatechin standards (Sigma Aldrich; St. Louis, MO, USA) were added sequentially to the 96-well plate, followed by adding 100 μL of 4-dimethyl-amino-cinnamaldehyde (DMACA, Sigma Aldrich, St. Louis, MO, USA) solution (1% DMACA (*w/v*) dissolved in 1:1 methanol/6N hydrochloric acid solution) to each well. The 96-well plate was placed in the microplate reader (SpectraMax 190, Molecular Devices, San Jose, CA, USA) and shaken for 5 min at room temperature. The absorbance values were measured at 640 nm for each well. The soluble PA content was quantified relative to the standard curve drawn from the (-)-epicatechin concentration and the corresponding absorbance values.

**Determination of Insoluble PA:** The precipitate obtained after the above extraction of the soluble PA crude extract was lyophilized for 24 h into the freeze-dried powder. The powder was resuspended with 1 mL of butanol/hydrochloric acid (95:5, *v/v*) solution, and sonicated in an ice water mixture for 30 min in darkness. The sonicated solution was incubated in a metal bath at 95 °C for 1 h, and then, it was cooled to room temperature and centrifuged at 4 °C for 5 min at 12,000× *g*. Finally, 100 μL of the reaction solution was transferred to a 96-well plate, and measured at 550 nm in the microplate reader. The insoluble PA content was quantified relative to the standard curve drawn from procyanidin B1 (PB1, (-)-epicatechin-(4β→8)-(+)-catechin, Sigma Aldrich, St. Louis, MO, USA) with final concentrations of 0, 50, 100, 150, 200, 400 μg/mL and the corresponding absorbance values.

### 2.2.3. Analysis of Flavan-3-ol Units

The flavan-3-ol units of soluble PA and insoluble PA were analyzed following the method of Yu et al. [29] with some modifications. An excess of nucleophilic solution, phloroglucinol lysis solution (50 g/L with 0.5% Vc), was used to completely hydrolyze the PA under acidic conditions, breaking the flavan bond and releasing the extension unit with phloroglucinol and the terminal unit without phloroglucinol. The flavan-3-ol units were identified qualitatively and quantitatively. The mDP values of soluble PA and insoluble PA were calculated separately.

**Derivatization of Soluble PA:** Soluble PA extract was lyophilized into dried powder. In addition to soluble PA, free flavan-3-ol monomers were included in the extract. For assessing free monomer content in the seeds, 100 μL extract was placed into 2 mL centrifuge tubes, lyophilized and redissolved in 200 μL of methanol (suitable for HPLC, ≥99.9%) solution (50%, methanol/water). The dissolved solution was used for the determination of free flavan-3-ol monomers in the seeds.

For assessing flavan-3-ol units of soluble PA, the dried powder obtained from 100 μL extract was mixed with 200 μL of phloroglucinol lysis solution, incubated for 20 min at 50 °C in the metal bath, blown by nitrogen until nearly completely dry, and resuspended in 200 μL of 50% methanol (suitable for HPLC, ≥99.9%) solution. The suspension was used to determine the terminal and extension monomers. Both the above solutions were filtered through a 0.22 μm membrane prior to instrumental detection.

**Derivatization of Insoluble PA:** The insoluble PA precipitate powder was added with 0.5 mL of phloroglucinol lysis solution, incubated for 20 min at 50 °C, and then centrifuged. The supernatant was transferred to a 5 mL centrifuge tube. The procedure above was repeated twice to enable full extraction. Then, 100 μL of the supernatant was blown by nitrogen until completely dry and added with 200 μL of 50% methanol (suitable for HPLC, ≥99.9%) solution, then filtered through a 0.22 μm membrane. The solution was used to determine terminal and extension monomers of insoluble PA.

**Detection of flavan-3-ols by HPLC-QqQ-MS/MS:** The flavanol units in the above soluble and insoluble PA samples were analyzed by an Agilent 1200 series high-performance liquid chromatography triple quadrupole mass spectrometer (HPLC-QqQ-MS/MS) [29].

The injection volume was 5 μL. The instrument parameters were set as follows: chromatographic column was a Poroshell 120 EC-C18 (150 × 2.1 mm, 2.7 μm, Agilent Technologies, Palo Alto, CA, USA), the column temperature was 55 °C; mobile phase A was water solution containing 0.1% formic acid; mobile phase B was a 1:1 methanolic acetonitrile solution (*v*/*v*) containing 0.1% formic acid. The mobile phase elution program was as follows: 0–14 min at 10–28% (B), 14–15 min at 28–10% (B), 15–17 min maintained at 10% (B) with a mobile phase flow rate of 0.400 mL/min. At the end of the 17 min pre-run, the post-run was started. That is, the 10% B mobile phase was restored to flush the column for 2 min and then equilibrated. The mass spectrometry used an electrospray ion source with electrospray voltage setting at ±4 kV according to the ionization characteristics of the flavan-3-ol. The ion source temperature was 150 °C. The nebulizing gas was high-purity nitrogen, the gas pressure was 35 psi, the gas flow was 10 L/min, and gas temperature was 350 °C. The detector was monitored by multiple reaction monitoring (MRM).

**Qualitative and Quantitative Experiments of Flavan-3-ol and Calculation of mDP:** The content of free monomers and terminal units in soluble and insoluble PA in seeds was quantified by C, EC, ECG standards' peak time of the characteristic ion and the standard curve drawn from the concentration and peak area of the standard solution. The content of the extension units was quantified by the concentration of the standard PB1 after derivatization under the same conditions as the samples and the corresponding standard curve; the unit was expressed of mg/g. seed. The mDP of the soluble/insoluble PA fraction was calculated separately using the following equations [30].

$$\text{mDP (Soluble PA)} = \frac{\textbf{PA units}^{-\textbf{P}} + \textbf{PA units} - \textbf{free monomers}}{\textbf{PA units} - \textbf{free monomers}}$$

$$\text{mDP (Insoluble PA)} = \frac{\textbf{PA units}^{-\textbf{P}} + \textbf{PA units}}{\textbf{PA units}}$$

Note: −P refers to the flavan-3-ol unit that is added to the nucleophile.

### 2.3. Data Processing and Statistical Analysis

Data were organized by Microsoft Office 2019 (Microsoft, Redmond, WA, USA), IBM SPSS Statistics 21 (IBM, New York, NY, USA) and R 4.1.2 (R Core Team) for two-way ANOVA (Duncan's, $p < 0.05$) and independent samples t tests. They were plotted using GraphPad Prism 8.4.0 (GraphPad Software; San Diego, CA, USA), Origin 2022 (OriginLab; Northampton, MA, USA) and MATLAB 7.0 (MathWorks, Natick, MA, USA).

## 3. Results

### 3.1. Differences in the Ripening Process of Grapes

The changes of total soluble solids (TSS) and titratable acid (TA) contents in the 'Cabernet sauvignon' and 'Marselan' grape berries harvested from four sub-regions are shown in Figure 2.

For 'Cabernet sauvignon', the grape berries from the HB, in comparison to the other three sub-regions, always had significantly higher TSS content. After full coloration, berry TSS accumulation speed was also faster, reaching 27.3 °Brix at the 30 days after full coloration (DAFC). In contrast, the 'Cabernet sauvignon' grapes from both ZH and YQ were characterized by the slow speed of TSS accumulation, which respectively came up to 25.5 and 24.1 °Brix at 50 DAFC. At commercial harvest, the 'Cabernet sauvignon' berries of YQ had the highest TA content at 19.54 g/L, while this variety of HB had the lowest TA content at 12.10 g/L. These results indicated that 'Cabernet sauvignon' berries of YQ had the slowest ripening speed among the studied sub-regions. Similarly, 'Marselan' of YQ berries also presented the slowest ripening process, only reaching 24.6 °Brix at 50 DAFC. Meanwhile, the TA contents of 'Marselan' berries were higher in YQ than of those in the other two regions, except for ZH. In contrast, the speed of TSS accumulation was higher in ZH and HB than that in the other two sub-regions.

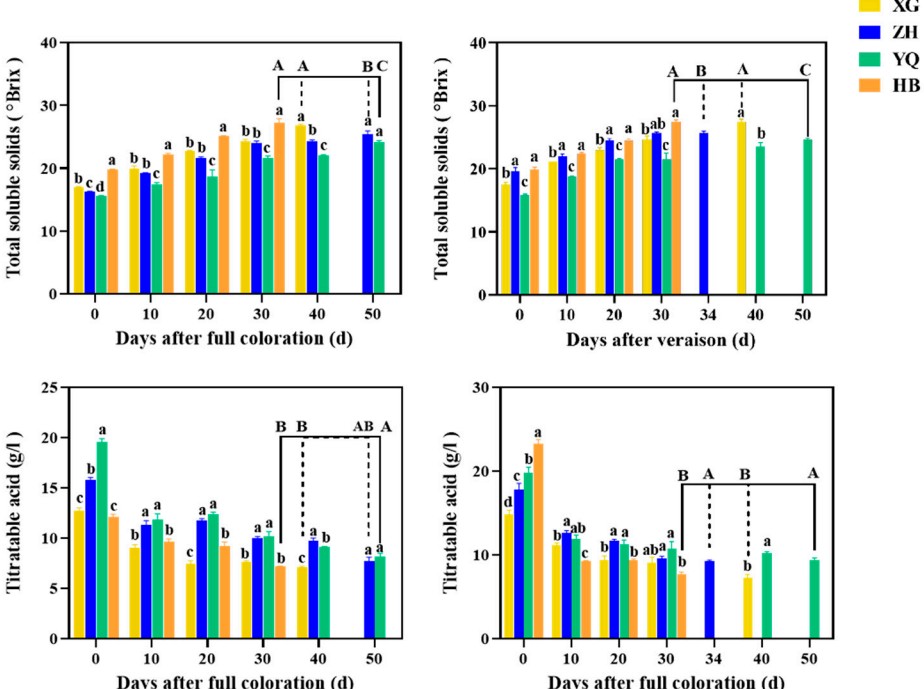

**Figure 2.** Changes of total soluble solids (TSS) and titratable acid (TA) content during ripening in 'Cabernet sauvignon' (**left**) and 'Marselan' (**right**) grape berries from four sub-regions. Different lowercase letters indicate significant differences between the samples from different sub-regions at the indicated time. Different capital letters indicate significant differences between berries from different sub-regions at the commercial harvest (Duncan's test, $p < 0.05$).

From Figure 3, it can be seen that both 'Cabernet sauvignon' and 'Marselan' berries in YQ took more than 120 days from full flowering to ripening harvest, longer than in the other three sub-regions. Combined with the variation in TSS and TA mentioned above, it was considered that the YQ berry ripening process was the slowest. In contrast, it took only 15 days to complete berry coloration for both varieties of HB and 30 days to reach commercial harvest from 100% véraison, showing a rapid ripening progression. The 'Cabernet sauvignon' grapes from ZH and YQ took 50 days from full coloration to commercial harvest, indicating a slow speed of sugar accumulation in berries.

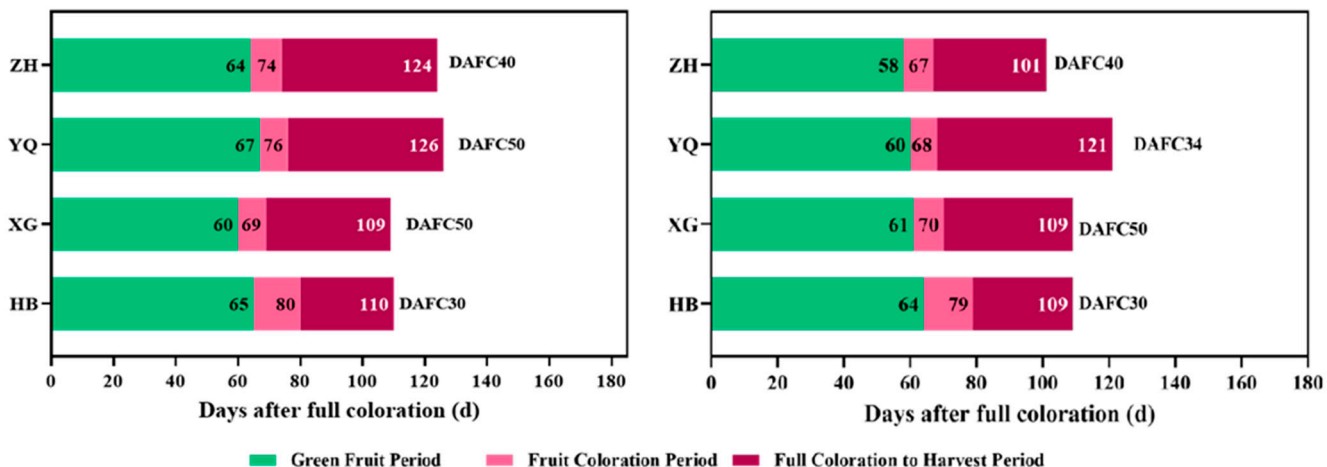

**Figure 3.** Fruit ripening process of 'Cabernet sauvignon' (**left**) and 'Marselan' (**right**) in four sub-regions. The numbers in the bars represent days after full flowering at the end of each stage.

For 'Marselan' berries, the TSS content of berries in ZH increased rapidly and was significantly higher than that of the grapes of XG and YQ, except for the time of DAFC30 and commercial harvest. Although the YQ 'Marselan' berries had a long growth period, they took only 8 days to accomplish the process from the beginning to complete coloration. These results indicates that sugar accumulation is not synchronized with phenol maturation.

### 3.2. Differences in Soluble and Insoluble PA Accumulation in Seeds

As a whole, soluble PA exhibited a decreasing followed by an increasing trend in the seeds, but the trough appeared at different times for the same varieties planted in the four sub-regions (Figure 4). The 'Cabernet sauvignon' and 'Marselan' grapes of XG were commercially harvested (13 September) earlier than those of the other three sub-regions, and the soluble PA content of the seeds decreased to a minimum at 20 DAFC, followed by an increase to a relatively high content at the commercial harvest. In contrary, the HB grape berries ripened faster, and the soluble PA content of 'Cabernet sauvignon' seeds was significantly higher than those of the other sub-regions in all the studied periods, while the soluble PA content of 'Marselan' seeds also had a high content all the time. Although the grape berries of YQ, in comparison to the other three sub-regions, ripened more slowly, the soluble PA of the seeds continued to accumulate after declining to the trough and ultimately reached the same level as that of the other three sub-regions. The ZH 'Marselan' were harvested the earliest on 8 September (see Table 2), and the seed soluble PA was the lowest among the samples from the commercial harvest period (Figure 4).

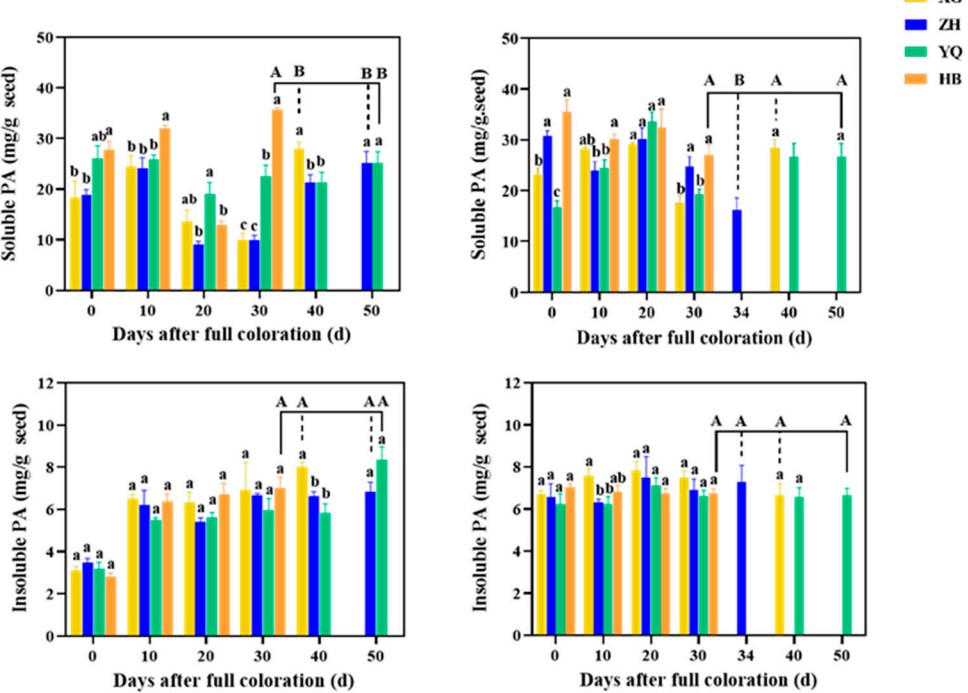

**Figure 4.** PA content of four different sub-regions of 'Cabernet sauvignon' (**left**) and 'Marselan' (**right**). Explanation about the lowercase and uppercase letters is the same as that in Figure 2.

Except for 'Cabernet sauvignon' of DAFC 0, the insoluble PA in the seeds of both varieties were always kept at a relatively stable level in the ripening process, with no significant difference presented among the samples from commercial harvest period. In the ripening process, although the 'Cabernet sauvignon' seeds of DAFC 40 and 'Marselan' seeds of DAFC 10 from ZH and YQ presented lower levels of insoluble PA, at the other specific sampling points, no difference in the insoluble PA content was presented between the seeds from different sub-regions (Figure 4).

### 3.3. Differences in Soluble and Insoluble PA Composition Units and Their Contents

The composition units of soluble PA and insoluble PA and their respective content were analyzed in the grape seeds of the commercial harvest period. In addition, we detected free flavan-3-ol monomers in the seeds. Three flavan-3-ol units, C, EC and ECG, were identified (Table 3). In terms of free monomers, the content of C was the highest, while the content of ECG was the lowest. For either 'Cabernet sauvignon' or 'Marselan', the HB seeds had the highest content of C, EC and ECG (Table 3). The possibility is thus inferred that the short ripening duration of HB grape berries was not conducive to flavan-3-ol monomers condensing into PA.

**Table 3.** The composition and content of soluble PA and insoluble PA in the seeds from four sub-regions (mg/g. seed).

| PA Composition | 'Cabernet Sauvignon' | | | | 'Marselan' | | | |
|---|---|---|---|---|---|---|---|---|
| | ZH | YQ | XG | HB | ZH | YQ | XG | HB |
| **Free monomers** | | | | | | | | |
| (+)-Catechin | 2.19 ± 0.14 [b] | 2.02 ± 0.23 [b] | 2.07 ± 0.11 [b] | 3.06 ± 0.32 [a] | 1.89 ± 0.12 [c] | 2.01 ± 0.03 [c] | 2.32 ± 0.16 [b] | 2.86 ± 0.07 [a] |
| (-)-Epicatechin | 1.28 ± 0.09 [b] | 1.25 ± 0.12 [b] | 1.19 ± 0.05 [b] | 1.73 ± 0.13 [a] | 1.14 ± 0.09 [b] | 1.14 ± 0.07 [b] | 1.29 ± 0.15 [ab] | 1.42 ± 0.18 [a] |
| (-)-Epicatechin-3-gallate | 0.16 ± 0.01 [b] | 0.17 ± 0.01 [b] | 0.14 ± 0.01 [b] | 0.27 ± 0.03 [a] | 0.25 ± 0.02 [b] | 0.29 ± 0.02 [b] | 0.25 ± 0.02 [b] | 0.38 ± 0.02 [a] |
| Total | 3.63 ± 0.25 [b] | 3.44 ± 0.36 [b] | 3.40 ± 0.16 [b] | 5.06 ± 0.44 [a] | 3.28 ± 0.14 [c] | 3.44 ± 0.03 [c] | 3.87 ± 0.33 [b] | 4.66 ± 0.27 [a] |
| **Soluble PA** | | | | | | | | |
| Terminal units | | | | | | | | |
| (+)-Catechin | 2.84 ± 0.17 [bc] | 2.63 ± 0.22 [c] | 3.14 ± 0.14 [ab] | 3.20 ± 0.16 [a] | 2.05 ± 0.17 [a] | 1.80 ± 0.76 [a] | 2.41 ± 0.23 [a] | 2.33 ± 0.18 [a] |
| (-)-Epicatechin | 1.61 ± 0.10 [b] | 1.67 ± 0.09 [ab] | 1.53 ± 0.13 [b] | 1.87 ± 0.09 [a] | 1.05 ± 0.13 [ab] | 0.77 ± 0.34 [b] | 1.23 ± 0.17 [a] | 0.98 ± 0.12 [ab] |
| (-)-Epicatechin-3-gallate | 3.29 ± 0.22 [b] | 2.92 ± 0.16 [c] | 3.3 ± 0.10 [b] | 4.19 ± 0.22 [a] | 3.71 ± 0.14 [ab] | 3.10 ± 0.47 [b] | 3.57 ± 0.45 [ab] | 4.02 ± 0.35 [a] |
| Total | 7.73 ± 0.46 [b] | 7.23 ± 0.39 [b] | 7.97 ± 0.37 [b] | 9.26 ± 0.33 [a] | 6.80 ± 0.21 [a] | 5.67 ± 1.58 [a] | 7.21 ± 0.78 [a] | 7.33 ± 0.42 [a] |
| Extension units | | | | | | | | |
| (+)-Catechin | 1.19 ± 0.11 [a] | 1.09 ± 0.22 [a] | 1.02 ± 0.14 [a] | 1.46 ± 0.40 [a] | 1.54 ± 0.10 [a] | 1.27 ± 0.14 [a] | 1.39 ± 0.08 [a] | 1.51 ± 0.32 [a] |
| (-)-Epicatechin | 51.97 ± 1.51 [b] | 53.24 ± 0.47 [ab] | 52.01 ± 1.99 [b] | 55.1 ± 1.17 [a] | 53.61 ± 1.23 [a] | 47.99 ± 6.09 [a] | 53.76 ± 1.89 [a] | 52.33 ± 1.17 [a] |
| (-)-Epicatechin-3-gallate | 8.47 ± 0.28 [b] | 8.77 ± 0.21 [b] | 8.28 ± 0.43 [b] | 9.55 ± 0.18 [a] | 11.84 ± 0.17 [a] | 10.72 ± 1.36 [a] | 11.07 ± 0.60 [a] | 11.84 ± 0.42 [a] |
| Total | 61.64 ± 1.55 [b] | 63.1 ± 0.19 [ab] | 61.31 ± 2.55 [b] | 66.11 ± 1.55 [a] | 66.99 ± 1.31 [a] | 59.98 ± 7.58 [a] | 66.22 ± 2.56 [a] | 65.68 ± 1.89 [a] |
| Total flavanols | 73.00 ± 2.14 [b] | 73.77 ± 0.64 [b] | 72.68 ± 2.81 [b] | 80.44 ± 1.25 [a] | 77.08 ± 1.45 [a] | 69.09 ± 9.13 [a] | 77.30 ± 3.59 [a] | 77.66 ± 2.56 [a] |
| mDP | 7.31 ± 0.25 [b] | 7.86 ± 0.37 [a] | 7.06 ± 0.10 [bc] | 6.70 ± 0.33 [c] | 9.10 ± 0.32 [a] | 10.12 ± 2.06 [a] | 8.44 ± 0.51 [a] | 8.37 ± 0.20 [a] |
| **Insoluble PA** | | | | | | | | |
| Terminal units | | | | | | | | |
| (+)-Catechin | 0.21 ± 0.05 [b] | 0.22 ± 0.01 [b] | 0.23 ± 0.03 [b] | 0.38 ± 0.03 [a] | 0.14 ± 0.01 [ab] | 0.17 ± 0.02 [ab] | 0.15 ± 0.02 [b] | 0.22 ± 0.05 [a] |
| (-)-Epicatechin | 0.12 ± 0.03 [b] | 0.13 ± 0.01 [b] | 0.12 ± 0.02 [b] | 0.18 ± 0.01 [a] | 0.07 ± 0.01 [ab] | 0.09 ± 0.01 [ab] | 0.07 ± 0.01 [b] | 0.09 ± 0.02 [a] |
| (-)-Epicatechin-3-gallate | 0.14 ± 0.03 [b] | 0.14 ± 0.01 [b] | 0.14 ± 0.00 [ab] | 0.18 ± 0.02 [a] | 0.13 ± 0.01 [ab] | 0.13 ± 0.01 [ab] | 0.12 ± 0.01 [b] | 0.14 ± 0.01 [a] |
| Total | 0.47 ± 0.10 [b] | 0.49 ± 0.02 [b] | 0.49 ± 0.04 [b] | 0.73 ± 0.06 [a] | 0.34 ± 0.03 [b] | 0.39 ± 0.03 [ab] | 0.34 ± 0.04 [b] | 0.46 ± 0.08 [a] |
| Extension units | | | | | | | | |
| (+)-Catechin | 0.22 ± 0.04 [a] | 0.24 ± 0.04 [a] | 0.23 ± 0.01 [a] | 0.26 ± 0.07 [a] | 0.22 ± 0.01 [a] | 0.23 ± 0.02 [a] | 0.20 ± 0.03 [a] | 0.21 ± 0.01 [a] |
| (-)-Epicatechin | 6.17 ± 0.80 [a] | 7.15 ± 0.53 [a] | 6.34 ± 0.16 [a] | 7.17 ± 1.41 [a] | 6.91 ± 0.14 [a] | 6.82 ± 0.21 [a] | 6.21 ± 0.75 [a] | 6.49 ± 0.17 [a] |
| (-)-Epicatechin-3-gallate | 0.99 ± 0.19 [a] | 1.08 ± 0.08 [a] | 0.96 ± 0.03 [a] | 1.11 ± 0.28 [a] | 1.54 ± 0.04 [a] | 1.50 ± 0.09 [a] | 1.30 ± 0.18 [b] | 1.38 ± 0.04 [ab] |
| Total | 7.38 ± 1.02 [a] | 8.47 ± 0.56 [a] | 7.53 ± 0.19 [a] | 8.54 ± 1.71 [a] | 8.67 ± 0.18 [a] | 8.55 ± 0.27 [a] | 7.72 ± 0.96 [a] | 8.07 ± 0.20 [a] |
| Total flavanols | 7.85 ± 1.08 [a] | 8.96 ± 0.55 [a] | 8.02 ± 0.15 [a] | 9.27 ± 1.77 [a] | 9.01 ± 0.21 [a] | 8.93 ± 0.27 [a] | 8.05 ± 0.21 [a] | 8.53 ± 0.25 [a] |
| mDP | 13.03 ± 2.41 [a] | 14.11 ± 1.46 [a] | 12.66 ± 1.44 [ab] | 9.55 ± 1.15 [b] | 20.7 ± 1.41 [a] | 17.83 ± 1.55 [a] | 18.44 ± 0.47 [a] | 14.47 ± 2.34 [a] |

Note: Different letters indicate that the differences among different sub-regions for the same variety reached the level of significance (Duncan's test, $p < 0.05$) (same below).

With regard to soluble PA, their terminal unit was rich in C, followed by ECG, while the extension unit preferred EC, which was consistent with the results reported by Bautista et al. [31] and Kennedy et al. [32]. Among the 'Marselan' grape seeds of four sub-regions, the terminal unit C and three extension units all had no noticeable difference, except that soluble PA of YQ contained lower levels of terminal units EC and ECG. Moreover, terminal units EC and ECG had no significant differences among the other three sub-regions (Table 3). For 'Cabernet sauvignon' grapes, the content of each component in the terminal and extension units in the HB seeds was the highest, which is associated with the highest soluble PA (Figure 4). The total content of soluble PA composition units was between 69.06 and 80.44 mg/g. seed, which was much higher than insoluble PA (7.85–9.27 mg/g. seed), which corresponds to their respective condensates.

Unlike soluble PA, insoluble PA of the seeds had no preference to terminal unit C, EC and ECG, and the composition units showed similar contents. Relatively speaking, there was the higher content of terminal unit C, rather than ECG, which indicated that the galloylation rate of insoluble PA was lower than that of soluble PA. The insoluble PA of both ZH and YQ 'Cabernet sauvignon' seeds compared to HB contained a lower content of

terminal units. For 'Marselan', the seeds of XG had lower terminal unit content than those of HB. The extension units of insoluble PA were characterized by relatively rich EC, the content of which was higher than those of C and ECG. As to the content of extension units of insoluble PA, there was no obvious difference among the seeds from four sub-regions (Table 3).

The mDP of seed PA was calculated, and the results are listed in Table 3. The mDP values of 'Marselan' seed soluble PA and insoluble PA, respectively, were 8.37–10.12 and 14.47–20.70, which were higher than the corresponding mDP of 'Cabernet sauvignon' seeds. In addition, there were no significant differences among the sub-regions with regard to the mDP of 'Marselan' seed soluble and insoluble PA. However, for 'Cabernet sauvignon', the mDP values of both soluble and insoluble PA in the seeds of HB were the lowest. The HB grapes showed the shortest ripening duration (Figure 3) among the studied sub-regions. In comparison to HB, the seeds of YQ 'Cabernet sauvignon' had higher mDP values of soluble and insoluble PA (Table 3), and the grapes of YQ had longer ripening duration (Figure 3). The comprehensive above results indicate that a slow fruit ripening process may be profitable to the polymerization of flavan-3-ols, thus resulting in PA with a high mDP value.

We further compared the proportion of each flavan-3-ol unit to the total content in soluble and insoluble PA among the four sub-regions (Table 4). The results revealed that some differences were present among the seeds from the detected sub-regions, in particular, between HB and the other sub-regions. Given that ECG is closely associated with the extractability and astringency intensity of PA, the proportion of ECG content to the total was particularly concerned. It was found that this proportion in soluble PA of the HB seeds was significantly higher than that of the XQ seeds for both 'Cabernet sauvignon' and 'Marselan'. Meanwhile, the ECG proportion in HB 'Cabernet sauvignon' seed soluble PA was significantly higher than that in ZH and YQ. For the seed insoluble PA, the content proportion of ECG in the 'Marselan' seeds of HB was significantly lower than that in the XG.

**Table 4.** Percentage of flavan-3-ol monomers (%) in the PA composition units of seeds from different sub-regions.

| PA Composition | 'Cabernet Sauvignon' | | | | 'Marselan' | | | |
|---|---|---|---|---|---|---|---|---|
| | ZH | YQ | XG | HB | ZH | YQ | XG | HB |
| **Soluble PA** | | | | | | | | |
| (+)-Catechin | 5.81 ± 0.08 [a] | 5.29 ± 0.17 [b] | 6.01 ± 0.13 [a] | 6.18 ± 0.31 [a] | 4.88 ± 0.40 [a] | 4.60 ± 0.79 [a] | 5.17 ± 0.22 [a] | 5.25 ± 0.27 [a] |
| (-)-Epicatechin | 77.23 ± 0.45 [a] | 78.09 ± 0.34 [a] | 77.27 ± 0.22 [a] | 75.60 ± 0.68 [b] | 74.06 ± 0.46 [ab] | 74.32 ± 0.66 [a] | 74.91 ± 0.61 [a] | 73.03 ± 0.57 [b] |
| (-)-Epicatechin-3-gallate | 16.96 ± 0.46 [b] | 16.63 ± 0.50 [b] | 16.72 ± 0.12 [b] | 18.23 ± 0.40 [a] | 21.07 ± 0.07 [a] | 21.07 ± 0.30 [a] | 19.92 ± 0.50 [b] | 21.71 ± 0.43 [a] |
| **Insoluble PA** | | | | | | | | |
| (+)-Catechin | 5.54 ± 0.69 [b] | 5.14 ± 0.01 [b] | 5.69 ± 0.31 [ab] | 6.99 ± 0.87 [a] | 3.95 ± 0.17 [b] | 4.48 ± 0.25 [ab] | 4.40 ± 0.10 [a] | 5.04 ± 0.57 [ab] |
| (-)-Epicatechin | 80.21 ± 1.26 [ab] | 81.25 ± 0.92 [a] | 80.52 ± 0.49 [ab] | 79.24 ± 1.12 [b] | 77.51 ± 0.13 [b] | 77.27 ± 0.12 [b] | 78.02 ± 0.43 [a] | 77.15 ± 0.13 [b] |
| (-)-Epicatechin-3-gallate | 14.26 ± 0.79 [a] | 13.60 ± 0.52 [a] | 13.79 ± 0.32 [a] | 13.77 ± 0.85 [a] | 18.54 ± 0.12 [a] | 18.25 ± 0.35 [ab] | 17.58 ± 0.38 [b] | 17.81 ± 0.47 [b] |

Note: Different letters indicate that the differences among different sub-regions for the same variety reached the level of significance (Duncan's test, $p < 0.05$) (same below).

## 4. Discussion

The development of seeds begins with embryo formation and endosperm growth, and it is known that the genes responsible for the biosynthesis of PA in seeds start expression after fertilization. Researchers generally find that in grape seeds, flavan-3-ols continue accumulation before véraison, followed by a decline until ripening and harvest [33]. During the development of grape berries, the accumulation of flavan-3-ol monomers and polymers does not proceed simultaneously. The 90% polymers are formed before véraison and reach peak values and then decrease during fruit ripening to harvest, mainly owing to the oxidation of PA. At that time, seed coat browning happens while the seed PA tightly combines with the cell wall. The combination not only provides protection for the seed coat, but also limits the extraction efficiency of seed PA [34]. In this study, we found that

the content of seed soluble PA showed large fluctuations in the late stages of berry ripening for either 'Cabernet sauvignon' or 'Marselan', which is consistent with previous research results [35]. Researchers thought that during the late ripening stage, seeds begin to change from green to pale yellow to eventually brown [32]. Full browning is considered as a mark of seed ripening. The brown part is mainly insoluble PA bound to the cell wall [33]. Therefore, it is believed that the flavan-3-ol monomers continuously condense into PA as the berries ripen, with oxidative cross-linking of PA with the cell wall always increasing. The combined results allow soluble PA of the seeds to exhibit fluctuations.

The grape berry ripening process depends on the environmental conditions. Although the seeds are located in the inner space of grape berries and the influence of external factors may be indirect, the external factors could alter the ripening progression of grape berries and further impact the generation of PA. A common phenomenon is observed such that the accumulation of sugar in grape berry is not synchronized with seed browning. In this study, the time from full coloration to commercial harvest is the shortest, and the sugar accumulation speed of HB berries was the fastest. Correspondingly, there was highest content of soluble PA in the HB 'Cabernet sauvignon' seeds. Moreover, its composition unit contained the highest proportion of ECG. On the contrary, the 'Cabernet sauvignon' grapes of YQ had the longest ripening period, and sugar accumulation was the slowest. The mDP of both soluble and insoluble PA of the seeds was relatively high and significantly higher than that of HB, indicating that the slowing down of the ripening rate is beneficial to the conversion of PA into larger polymers and reduces the content of ECG in the PA composition units. Although different grape varieties exist with certain difference, it is still seen that the XG 'Marselan' grapes have a moderate ripening rate, and soluble and insoluble PA content as well as mDP value in the commercially harvested grape seeds are not significantly different from other sub-regions, whereas the ECG percentage in PA was significantly lower than other sub-regions. These also further illustrate that the slowing down of the ripening process lowers the proportion of ECG units in the PA.

Some reports have stated that the degree of polymerization and composition of PA are closely related to the intensity of bitterness and astringency. In general, with the increase in PA polymerization degree, the maximum intensity of astringency (Imax) increases, while the Imax and duration of bitterness decrease [36]. With the increase in EC and ECG proportion in the PA composition units, the intensity of astringency and bitterness will be enhanced [6]. Thus, it is expected that the seed PA of grapes, in particular extractable soluble PA, contains a relatively low ECG proportion. Based on these above results, we believe that the accelerated fruit ripening process is not conducive to the formation of good-tasting PA in seeds. It is required to take some effective viticulture measures to slow down the grape berry ripening process in dry-hot wine producing regions such as in western China and to cope with the challenge of global warming.

## 5. Conclusions

The study dissects the differences in the content and composition of soluble and insoluble PA in 'Cabernet sauvignon' and 'Marselan' seeds from four sub-regions. The main concern is on exploring the relationship between berry ripening progression and soluble and insoluble PA. This finding is of great significance for guiding the production of good wine grapes. The present results indicate that a fast sugar accumulation in ripening berries is accompanied by the formation of PA with a high galloylation proportion and low mDP in the seeds.

**Author Contributions:** Writing—original draft preparation, A.L.; chemical analysis and data analysis, J.W., A.L. and Q.S.; sampling, N.X., X.Y. and Q.S.; writing—review and editing, Q.P. and X.Y.; visualization, A.L. and J.W.; supervision, Q.P.; project administration, C.D. and Q.P. All authors have read and agreed to the published version of the manuscript.

**Funding:** This research was funded by the Key R&D projects in the Ningxia Hui Autonomous Region, grant number 2021BEF02014, to Q. P.; National Nature Science Foundation of China, grant number U20A2042, to C.D.

**Institutional Review Board Statement:** Not applicable.

**Informed Consent Statement:** Not applicable.

**Data Availability Statement:** The data used for the analysis in this study are available within the article.

**Acknowledgments:** The authors sincerely thank Chateau Yuanshi, Chateau Yuquan, Xige Estate and Hongbao Vineyard for grape sampling support. We also thank Lanxuan He from the Affiliated School of Peking University for assistance with the usage of GraphPad Prism 8.4.0, Origin 2022 and MATLAB 7.0.

**Conflicts of Interest:** The authors declare no conflict of interest.

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
