# Peer review of "Dissecting Seed Proanthocyandin Composition and Accumulation under Different Berry Ripening Process in Wine Grapes"

_horticulturae, doi:10.3390/horticulturae9010061_

Round 1

Reviewer 1 Report

The manuscript contains a valuable research study. However, some detailes should be improved:

1. English language should be carefully check. There are a lot of punctuation errors (lack of space between the words, e.g. lines 75, 84, 100, 105).

2. The Introduction section is too long.

3. In all Figures: the legend should be in the chronological order with the bars in the charts. Why, there is a bar at day 50 for 'Marselan' grape berries on the Figure 1 and 3,  if there was no sampling on day 50 (see Table 2)?

4. Line 310. Explain what is the modified system E-L? There is no information in the methodology.

5. Line 387. Should be "in the Table 3".

6. Line 394. Should be sub-regions instead sun-regions.

7. Conclusions should concern the results obtained. The last sentence should be explain in the context of results.

Reviewer 2 Report

I believe that the article is a complex one with a novelty very well underlined. The methodology is well described and the results well disscused.

Reviewer 3 Report

The study showed information about the seed tannin composition and accumulation during ripening process in four subzones in the Ningxia Region, China. Cabernet Sauvignon and Marselan grape varieties was used to study the changes in the composition of proanthocyanidins in seeds. The aims of the work are well defined, and the methodology used in the assay responds in good form to the objectives. The study has a major importance principally due that offers novel insights in the study of tannin composition, especially in the berry seeds and try to understand the accumulation of this phenolic compounds and the relationship with sugar accumulation. There are some issues that need revision.

Material and methods section

Lines 136-139. Are there differences in climatic parameters among the four subzones? Maybe is necessary show climatic parameters such as, humidity, maximum and minimum temperatures, solar radiation and also days degrees. Other aspect especially for the readers is the use of a figure (supporting information) that shows the geographical areas in a map.

Lines 139-140. Are the vines grafted? What is the clone used in each variety? Please explain as possible.

Lines 140-141. The authors need to explain the type of soils in terms of certain parameters such as soil depth, pH, fertility, presence of organic matter and drainage. Please explain as possible.

Line 162. Does full coloration mean 100% of veraison? Maybe is more academic talk about degrees of veraison instead coloration? Although in many papers the researcher uses DAV, days after veraison, the use un DAFC, days after full coloration could be used if the author indicate the percentage of veraison, so 100% veraison.

Line 165. What are the implications of no sampling in DAFC40 and DAFC50 especially in the HB zone? Please explain as possible

Line 267. The correct word is flavan

Discussion section

Line 444. The author says, “the accumulation of PA during…is influenced by environmental factors, such as regional differences” The use of this phrase for this study highlights the need for climate data of each subzone. Please consider for a better support of the conclusion obtained in this assay.

Reviewer 4 Report

Dear Authors and Editors

The article was written in a transparent manner. In its assumptions, the study is justified and interesting.

The topics presented in the article are appropriate for the Journal's profile.

The authors presented the discussed issues in a broad way.

The literature is selected in the right way.

The conclusions sum up the entire article appropriately.

However, I have a few technical notes:

- line 94 page 2 after "[16]." space,

- line 96 page 3 correct apostrophe in tannins,

- line 102 page 3 after "[18]." space,

- line 222 page 6 after "[28]" space,

- line 222 page 6 after "[28]" space,

- line 243 page 7 after "at 50 C" remove the space,

- line 276 page 7 after "=" space,

- line 345 page 9 after "period" space,

- line 437 page 13 after "brown" space,

- in Table 3 the presented values and their errors should have the same number of significant digits,In my opinion, this paper can be accepted for publication in Horticulturae after minor corrects.

Best regards
